



# Extreme floods in Europe: going beyond observations using reforecast ensemble pooling

Manuela I. Brunner[1] and Louise Slater[2]

[1]Institute of Earth and Environmental Sciences, University of Freiburg, Freiburg, Germany
[2]School of Geography and the Environment, University of Oxford, Oxford, United Kingdom

**Correspondence:** Manuela I. Brunner (manuela.brunner@hydrology.uni-freiburg.de)

**Abstract.** Assessing the rarity and magnitude of very extreme flood events occurring less than twice a century is challenging due to the lack of observations of such rare events. Here we develop a new approach, pooling reforecast ensemble members from the European Flood Awareness System (EFAS) to increase the sample size available to estimate the frequency of extreme local and regional flood events. We assess the added value of such pooling, determine where in Central Europe one might expect the most extreme events, and evaluate how event extremeness is related to physiographic and meteorological catchment characteristics. We work with a set of 234 catchments from the Global Runoff Data Center for which performance of simulated floods is satisfactory when compared to observed streamflow. We pool EFAS-simulated flood events for 10 perturbed ensemble members and lead times from 22 to 46 days, where flood events are only weakly dependent ($< 0.25$ average correlation across lead times). The resulting large ensemble (130 time series instead of one) enables analyses of very extreme events, which occur less than twice a century. We demonstrate that such ensemble pooling produces more robust estimates with considerably reduced uncertainty bounds (by $\sim 80\%$ on average) than observation-based estimates but may equally introduce biases arising from the simulated meteorology and hydrological model. Our results show that specific flood return levels are highest in steep and wet regions and are comparably low in regions with strong flow regulation through dams. Furthermore, our pooled flood estimates indicate that the probability of regional flooding is higher in Central Europe and Great Britain than in Scandinavia. We conclude that reforecast ensemble pooling is an efficient approach to increase sample size and to derive robust local and regional flood estimates in regions with sufficient hydrological model performance.

## 1 Introduction

Reliable estimates of the frequency and magnitude of extreme flood events are needed to develop suitable preparedness and adaptation measures. However, estimates of flood events occurring less than twice a century are usually affected by large uncertainty and low reliability due to the shortness of observed records. To increase the sample size available for flood frequency analysis, different model-based approaches have been proposed. These include stochastic simulations, where a statistical model is used to generate a large sample of flood events with similar characteristics as the observations (Keef et al., 2013; Quinn et al., 2019; Brunner et al., 2019; Brunner and Gilleland, 2020), or large climate ensembles (Deser et al., 2020) coupled with a hydrological model (van der Wiel et al., 2019), where a large ensemble of meteorological variables is fed into a hydrological





model to generate a streamflow time series ensemble. Another related and potentially valuable source of information is refore-
casts, i.e. forecasts run for past periods (Hamill et al., 2006). Extremes pooled from such reforecasts have been shown to have
considerable value for analysing rare events and estimating the frequency of different types of hydro-meteorological extremes
including extreme wind (Breivik et al., 2014; Osinski et al., 2016; Meucci et al., 2018), sea-surge levels (van den Brink et al.,
2004), wave heights (Breivik et al., 2013), precipitation (Thompson et al., 2017; Kelder et al., 2020), and the water balance
(van den Brink et al., 2005). Such reforecast pooling relies on reforecasts of the variable of interest and pools extreme events
extracted from ensemble members of different lead times and/or perturbed ensemble members generated by perturbing initial
conditions. Reforecast pooling is also referred to as the UNprecedented Simulated Extreme ENsemble (UNSEEN) approach
(Thompson et al., 2017; Kelder et al., 2020) because it enables the study of unprecedented simulated extremes absent in short
observational records. The approach relies on the limited predictive skill of medium-range (re)forecasts related to the rapid
growth of errors with increasing lead time (Hamill et al., 2006). At long lead times (>10 days), (re)forecasts of meteorological
variables such as wind or precipitation represent independent simulations. Despite the requirement of low (re)forecast skill, the
simulated distribution must still resemble the observed distribution in order to yield realistic design quantiles (Breivik et al.,
2014).

While this ensemble pooling or UNSEEN approach has been successfully used to assess the frequency of rare wind, wave
height, storm surge, and precipitation events (Breivik et al., 2014; Osinski et al., 2016; Meucci et al., 2018; Breivik et al., 2013;
van den Brink et al., 2004; Kelder et al., 2020), its potential value has not yet been assessed for flood frequency analyses. We
here propose to pool perturbed members from the European Flood Awareness System (EFAS) at different lead times to generate
a large ensemble of extreme flood events. We use this ensemble to: (1) assess how well the pooled ensemble method works
in different locations, and evaluate the conditions in which it improves flood frequency estimates, relative to observations; and
(2) determine the frequency of occurrence (return periods) of extreme and widespread flood events across Europe.

## 2  Methods and Materials

Our evaluation of the ensemble pooling approach for flood frequency analyses in Europe relies on a set of 234 catchments in
Central Europe with areas ranging from a first quartile of 698 km$^2$ to a third quartile of 11510 km$^2$ (min: 16 km$^2$, max: 159300
km$^2$, inter-quartile range: 10812 km$^2$) and elevations ranging from a first quartile of 35 m.a.s.l. to a third quartile of 309 m.a.s.l
(min: 2 m.a.s.l, max: 1852 m.a.s.l, inter-quartile range: 273 m.a.s.l) (Figure 1). The catchments selected for the analysis have
to fulfill three criteria: (1) observed streamflow must be available through the Global Runoff Data Centre (GRDC; The Global
Runoff Data Centre 56068 Koblenz Germany, 2019) for model evaluation; (2) catchments must be included in the Global
Streamflow Indices and Metadata Archive (GSIM; Do et al., 2018), which provides catchment boundaries and characteristics;
and (3) sites must show acceptable hydrological model performance in terms of high flows when comparing observed flows to
streamflow simulations derived through the European Flood Awareness System (EFAS; Barnard et al., 2020) using historical
climatology. Using this dataset of 234 European catchments, we first assess under which circumstances the ensemble pooling
approach is suitable when using hydrological simulations. Second, we evaluate where and how the use of a pooled ensemble





approach can benefit flood frequency analysis in terms of both best estimates and uncertainty. Finally, we derive local and regional flood estimates for Central and Northern Europe where the majority of our catchments are located.

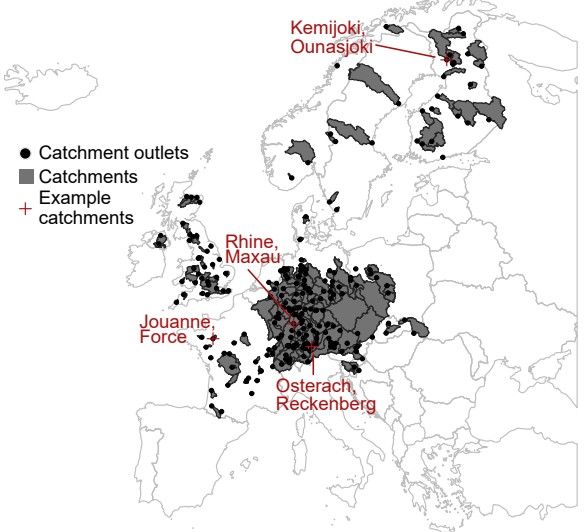

**Figure 1.** 234 catchments in Central Europe selected for the pooled frequency analysis based on model performance and availability of catchment boundaries and characteristics. Four example catchments used for illustration purposes are highlighted in red: (a) Kemijoki, Ounasjoki (strong summer flood regime), (b) Osterach, Reckenberg (summer flood regime), (c) Rhine, Maxau (winter flood regime), and (d) Jouanne, Force (strong winter flood regime).

## 2.1  Data

The pooled frequency analysis relies on reforecasts of daily streamflow time series generated through the European Flood Awareness System (EFAS). EFAS provides deterministic and probabilistic medium-range streamflow forecasts and early warning information (Bartholmes et al., 2009; Smith et al., 2016). It relies on numerical weather predictions from the European Centre of Medium-range Weather Forecasts (ECMWF), initial conditions derived using observed meteorological data, and the hydrological model LISFLOOD. LISFLOOD is a spatially distributed hydrological rainfall-runoff model based on Geographic Information Systems (GIS) developed by the Joint Research Centre (JRC) for operational flood forecasting at the pan-European scale (Thielen et al., 2009). It computes a water balance at a 6-hourly or daily time step for each grid cell (5km × 5km) using meteorological forcing data (precipitation, temperature, potential evapotranspiration, and evaporation rates), representing a variety of processes (snowmelt, soil freezing, surface runoff, soil infiltration, preferential flow, soil moisture redistribution, drainage to the groundwater system, groundwater storage, and groundwater base flow), and routing runoff produced for each grid cell through the river network using a kinematic wave approach. LISFLOOD has been calibrated using the Nash–Sutcliffe efficiency metric (Nash and Sutcliffe, 1970) for 693 catchments for which data were available through the GRDC for the period 1994–2002 (Smith et al., 2016).





In addition to streamflow forecasts, EFAS provides streamflow reforecasts generated by forcing LISFLOOD with medium-
to sub-seasonal range meteorological reforecasts (Barnard et al., 2020). Reforecasts are forecasts run for past periods (Hamill
et al., 2006). The EFAS reforecasts cover the 20-year period from 1999–2018 and are initialized twice a week on Mondays and
Thursdays with lead times of up to 46 days at a 6-hourly time step. They are driven with ensemble meteorological reforecasts
from ECMWF's numerical weather prediction model. The meteorological ensemble consists of 10 perturbed ensemble runs,
which were derived using the same numerical weather prediction model but varying initial conditions. Ensemble (re)forecasts
are often used to express the uncertainty of a (re)forecast by attributing probabilities to different (re)forecast values. We focus
our analysis on a subset of available lead times ($l_t$), i.e. we chose every 8th lead time available: $l_t = 0, 48, 96, ..., 1104$ hours.

Besides these streamflow reforecasts, EFAS also provides historical streamflow simulations driven with observed meteo-
rology (Mazzetti et al., 2019). We downloaded both EFAS reforecasts and historical runs from the Copernicus data store for
ensemble pooling and model evaluation, respectively. The EFAS simulations require some cleaning due to the presence of neg-
ative and extremely high implausible values (e.g. $2.5 * 10^{29}$). We set negative values to 0 and removed errors using an outlier
definition of $Q_{75} + 10 * R_{IQ}$, where $R_{IQ}$ is the inter-quartile range.

In order to identify catchments where EFAS produces reliable simulations, we also obtained observed daily streamflow for
all European catchments included in the Global Runoff Data Centre (GRDC) database (The Global Runoff Data Centre 56068
Koblenz Germany, 2019). We only retained those 1586 catchments for which data are available for the period 1991–2012.
For each catchment in the final selection, we obtained the catchment boundary and topographical characteristics from the
Global Streamflow Indices and Metadata Archive (GSIM; Do et al., 2018). We used the catchment boundaries to compute
annual mean areal hydro-meteorological characteristics including precipitation, temperature, evapotranspiration, snow-water-
equivalent, snowmelt, and soil moisture (sum over 4 layers) from the gridded ERA5-Land data set (ECMWF, 2019). For
illustration purposes, we chose 4 example catchments with different flood seasonalities as illustrated in Figure 1: (a) Kemijoki,
Ounasjoki (strong summer flood regime), (b) Osterach, Reckenberg (summer flood regime), (c) Rhine, Maxau (winter flood
regime), and (d) Jouanne, Force (strong winter flood regime).

## 2.2 Model and ensemble evaluation

We assess the quality of EFAS's historical streamflow simulations (generated with observed meteorological data) relative to
observed streamflow for all GRDC stations with observations for the period 1991–2012. The evaluation focuses on the period
1999–2012 as simulations only start in 1999 and observations are available until 2012. We compute different metrics which
focus on high flows, including the Kling–Gupta efficiency metric ($E_{KG}$; Gupta et al. (2009)) and the relative errors between
simulated and observed 95% ($Q_{95}$), 99% and 99.5% quantiles. High-flow simulation performance varies widely among the
1586 catchments. As reliable simulation of high flows is a prerequisite for flood frequency analyses, we only retain catchments
with 'satisfactory' performance in terms of $E_{KG}$ and $Q_{95}$. That is, we only choose stations with $E_{KG} > 0.6$ and relative $Q_{95}$
errors $< 10\%$. Only $\sim 15\%$ (234 out of the 1586) of all stations fulfill these criteria (Figure 1). The 234 catchments fulfilling
these criteria are retained for further analysis.





Next, we assess the suitability of the perturbed ensemble streamflow simulations for ensemble pooling by evaluating whether individual simulation runs can be considered independent and whether simulated distributions are stable across lead times, i.e. if there is model stability (Kelder et al., 2020). Ensemble member independence is an important factor determining the increase

in effective sample size achieved through ensemble pooling. If all $x$ simulation runs are independent, pooling increases effective sample size by $x$ times. However, if the $x$ simulation runs show a higher degree of dependence, pooling increases sample size by $y < x$ times. To assess (in)dependence between ensemble members in terms of flood magnitudes, we extract annual maxima (AM) flood events from each of the simulated ensemble time series (10 perturbed runs for each lead time). We subsequently calculate Spearman's rank correlation using the pairs of annual maxima time series (following Kelder et al., 2020). Note that

such correlation can only be computed for AM and not directly for peak-over-threshold (POT) series because POT series may differ in the number of events chosen for analysis and not just in timing and magnitude. It can therefore be assumed that the POT events used in our subsequent analyses are more independent than AM events. To illustrate this, we compute Spearman's correlation for POT time series, where non-exceedances were replaced by 0 (not ideal because this might artificially introduce some sort of dependence). Besides member independence, we assessed model stability, i.e. whether the generated ensemble

exhibits any changes in distribution with lead time. Ideally a pooled ensemble should not exhibit any drift. Such stability is assessed by comparing the distribution of AM events across different lead times. To evaluate model stability for a large number of catchments, we here quantify the dependence of simulated 95% flood quantiles (20-year return period) on lead time using Spearman's correlation.

A simulated streamflow time series can be biased because of uncertainties introduced through the modeling process. Substan-

tial bias indicates low model fidelity because there is limited agreement between observed and modeled distributions (DelSole and Shukla, 2010). Potential uncertainty sources introducing bias include meteorological input uncertainty due to the use of a numerical weather prediction system and hydrological parameter and model uncertainties. Any such biases must be corrected in order to align the simulated streamflow distributions with observed distributions. To do so, we apply non-parametric quantile mapping to the daily simulated discharge series using the R-package *qmap* (Gudmundsson, 2016), which has been found to

be more flexible and suitable than parametric mapping approaches (Gudmundsson et al., 2012). The values of the empirical cumulative distribution function of the observed and simulated time series are estimated for regularly spaced quantiles for the period 1999–2011, for which both simulations and observations are available. Then, quantile mapping is applied to the simulated distributions of the whole period 1999–2018. We also tested another commonly applied bias correction procedure which maps the simulated distribution using the mean bias ratio on extracted extremes (as done in other ensemble pooling

studies e.g. by Kelder et al. (2020)). However, we find that a correction by the mean bias ratio will still lead to biased flood estimates especially for long return periods. A comparison of the cumulative distribution functions of observed and simulated POT events shows that both non-parametric quantile mapping and correction by the mean bias ratio lead to an alignment of simulated with observed distributions and that quantile mapping produces more satisfactory results than correction by the mean bias ratio (Figure 2). We therefore use the non-parametrically quantile mapped series for the pooled flood frequency analysis.





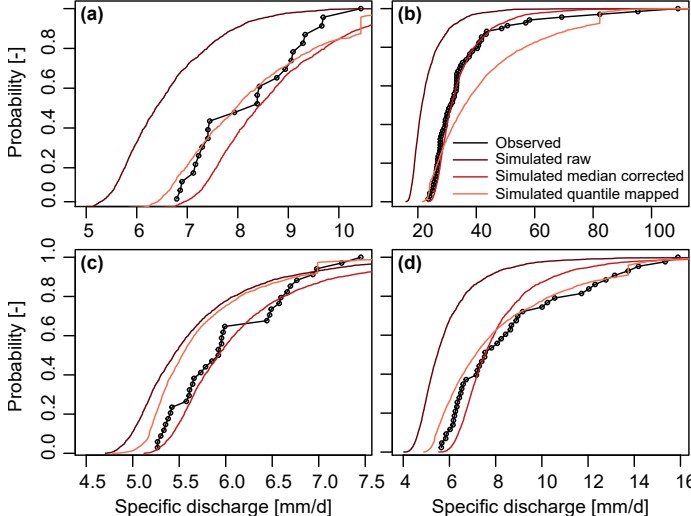

**Figure 2.** Comparison of observed and simulated cumulative distribution functions of POT flood events derived from: observed stream-flow time series, raw simulations without any bias correction, POT events corrected by the median ratio between observed and simulated POT distributions, and empirically quantile mapped simulations for the four example catchments: (a) Kemijoki, Ounasjoki, (b) Osterach, Reckenberg, (c) Rhine, Maxau, and (d) Jouanne, Force.

## 2.3 Frequency analysis

We define flood events using a peak-over-threshold (POT) approach to ensure inclusion of relevant events and to reduce the dependence between ensemble members (i.e. runs for different lead times and for different perturbations) compared to using AM events. The threshold is set to the 99th percentile and independence between events is ensured by prescribing a minimum time window of 10 days between events (Diederen et al., 2019; Brunner et al., 2020a, b). The POT approach is applied to each of the simulated time series generated for different lead times (24) and perturbations (10), i.e. for 240 time series per catchment.

For the local (catchment-specific) frequency analysis, we pool all POT events from the perturbed members of the lead times that can be considered to be independent, i.e. lead times $\geq 528$ hours or 22 days (see Section 3.2, Figure 3), which increases the sample size available for frequency analysis from $1 \times \sim 20$ events (roughly 1 event chosen per year on average) to 13 lead times $\times 10$ members $\times 20$ events $= 2600$ events.







**Figure 3.** Illustration of event pooling procedure across lead times $\geq 528$ hours and perturbed ensemble members.

We fit a theoretical Generalized Pareto distribution (GPD; Coles, 2001) to observed and pooled POT samples using maximum likelihood estimation. We use the fitted distributions to derive best estimate observed and simulated flood frequency curves, respectively, using probabilities corresponding to return periods between 1 and 200 years. In addition, we derive 90% confidence intervals for the observed and simulated frequency curves using bootstrapping, i.e. we draw $n = 1000$ random samples from the observed and simulated samples, respectively, to derive 1000 theoretical flood frequency curves. We then use

these resampled frequency curved to compute 90% confidence intervals for the estimated flood frequency curves. We compare simulated to observed flood quantiles corresponding to return periods of $T = 5, 10, 20, 50, 100$, and 200 years by computing relative differences between simulated and observed quantiles. We furthermore compare the uncertainty of these estimates by computing the relative difference in the range between the 95% and 5% quantile of the 1000 resampled estimates for each return period.

As a reference for these theoretical estimates, we provide empirical return period estimates of the observed flood events derived using the Weibull plotting position $T_{\mathrm{wb}} = m/(N + 1)$, where $N$ is the total number of events and $m$ is the rank of an event within the sample. To also represent the uncertainty of these empirical estimates, we perform another bootstrap experiment, which derives plotting positions for different samples where 1 year is removed at a time.

Next, we map flood quantiles estimated for return periods of $T = 10, 20, 50, 100$, and 200 years for the 234 catchments with

satisfactory model performance using the GPD distributions fitted to the pooled POT samples. To identify physiographical and hydro-meteorological characteristics important for explaining flood quantiles at different return periods, we use linear modeling. We fit different linear regression models of different size, i.e. with different numbers of explanatory variables, using exhaustive search (James et al., 2013) to predict flood quantiles with a specific return period, e.g. 10 years, using a set of explanatory variables frequently used to explain flood characteristics including altitude, latitude, longitude, catchment area,

number of dams in catchment, mean slope, population count, mean temperature, mean precipitation, mean evapotranspiration, mean SWE, mean snowmelt, and mean soil moisture as potential explanatory variables. Among the fitted models with different numbers of predictors, we identify the model with the smallest Bayesian Information Criterion (BIC) value for each return period and look at the explanatory variables retained in these models. The sign and magnitude of the retained regression coefficients will tell us something about the importance of each predictor in explaining flood quantiles for different return

periods.





After performing the local frequency analysis, we look at probabilities of regional flooding, i.e. estimate the return periods of events that affect a certain percentage of catchments within a larger region, i.e. river basin. We focus on the major river basins in Europe (HydroSHEDS; Lehner and Grill, 2013) and ask what is the probability that 30%, 50% and 70% of the catchments in each of these river basins are jointly affected by flooding, respectively. To compute such regional flooding probabilities, we

follow the the regional hazard estimation procedure proposed by Brunner et al. (2020b). For each region, i.e. large river basin, we (1) determine the available catchments located within the given region and focus on river basins with at least 5 catchments (out of the 234); (2) identify the number of flood events during which $p\%$ of the catchments are jointly flooded using a binary flood event matrix, which indicates for each catchment whether it was affected by specific flood events identified across all catchments, lead times, and perturbed members; (3) compute the probability of regional flooding using the Weibull plotting

position given by

$$p \quad (\%) = 100 \quad (x/(n+1)), \tag{1}$$

where $n$ is the total number of events affecting at least one of the stations in the region, and $x$ is the number of events where $r\%$ of the stations were affected.

## 3 Results

### 3.1 Model evaluation


The comparison of historical EFAS simulations with observed GRDC streamflow in $\sim 1600$ European catchments shows large variations in model performance with respect to both mean and high flows. Relative errors in mean discharge range from a first quartile of -15% to an upper quartile of +55%, those in the 95th quantile from -20% to +50%, those in the 99% and the 99.5% quantile from -30% to +40%. Kling-Gupta efficiency values range from a first quartile of -0.288 to an upper quartile

of 0.68. That is, independent of the quantile, over- or underestimation of flows can be substantial and not all of the $\sim 1600$ catchments show satisfactory model performance for flood frequency analysis. As model performance seems to be independent of geographic location, we select a set of 'quality' stations where model performance for high flows is considered satisfactory. To be selected for the flood frequency analysis, catchments had to have $E_{\mathrm{KG}} > 0.6$ and a relative $Q_{95}$ error $< 10\%$. Catchment selection is to some degree dependent on threshold choice with slightly more catchments being included if the threshold is set

on $Q_{95}$ instead of $Q_{99}$ and if lower percentage and $E_{\mathrm{KG}}$ thresholds are chosen. 234 out of the $\sim 1600$ sites (roughly 15% of initial sample size) fulfill both selection criteria and are retained for further analysis.

### 3.2 Ensemble evaluation

After identifying catchments with satisfactory model performance in terms of the EFAS historical runs, we assess the suitability of the streamflow ensemble generated using the perturbed numerical weather predictions and different lead times for ensemble

pooling. This assessment focuses on AM instead of POT flood samples because POT event identification can lead to the selection of an unequal number of events across lead times, which makes it impossible to compute correlations. We first





consider stability (i.e. lack of drift across lead times) of annual maxima flood events simulated for 24 lead times ranging from 0 to 46 days for one example catchment (Figure 4). Within each year, flood magnitudes don't differ systematically across lead times (Figure 4a) and cumulative flood distributions seem to be stable across lead times (Figure 4b).

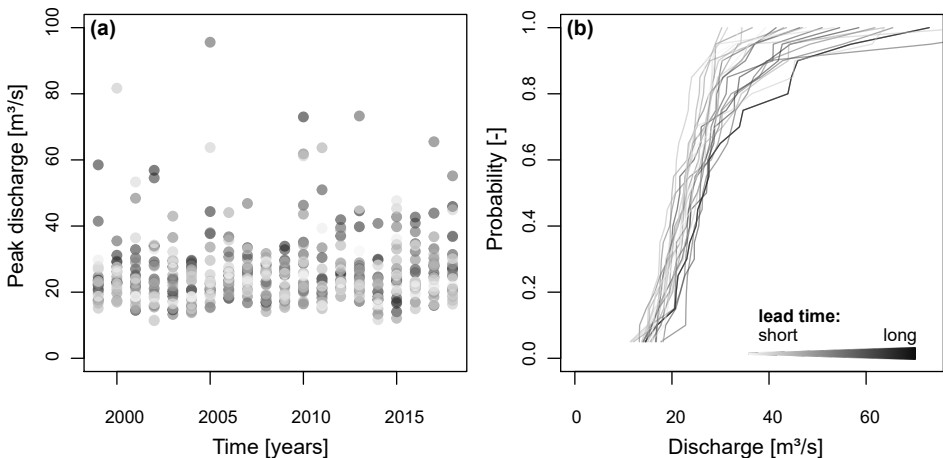

**Figure 4.** Stability across lead times from 0 to 46 days for one example catchment: (a) AM flood events extracted from streamflow time series simulated for 24 lead times (1 dot per lead time) and (b) cumulative distribution functions of AM flood samples per lead time (1 line per lead time). The darker the color, the longer the lead time.

We take a closer look at model stability for all catchments by assessing the dependence of the empirical 95% flood quantile on lead time using Pearson's correlation coefficient. Median correlation between lead time and the simulated 95% quantile across all catchments is 0.02 and the lower and upper quartiles are -0.32 and 0.35, respectively. That is, in most catchments, simulated flood quantiles are only weakly dependent on lead time, which suggests overall model stability. Some individual catchments may exhibit greater (positive or negative) forecast drift than others, and so researchers may wish to assess the

model stability more closely when working on individual case studies.

    We now take a look at AM (in)dependence across perturbed ensemble members by computing Spearman's rank correlation between pairs of AM series derived for the 10 perturbed ensemble members at each lead time. AM (in)dependence across perturbed ensemble members seems to depend both on the catchment and on lead time (Figure 5). Dependence is relatively high for short lead times and decreases with longer lead times. The strength of the dependence at $t = 0$, as well as the rate of

decrease with increasing lead time, depends on the catchment as illustrated by the different 'dependence decay' behavior of the four example catchments. While some catchments (e.g. b and d) show correlation values close to 0 for sufficiently long lead times, other catchments (e.g. a) show relatively high correlations of 0.5 even at lead times exceeding one month. That is, individual AM series for different ensemble members may not necessarily be fully independent in a hydrological context. This finding is in contrast to independence tests performed for other types of extremes such as extreme precipitation (Kelder et al.,

2020) or wind (Breivik et al., 2014), which found that simulated precipitation and wind extremes can be considered independent





after certain lead times. The residual dependence in the case of hydrological simulations is likely caused by the long memory of hydrological systems, which is related to storage processes e.g. in the soil or the cryosphere, and the persistence of the effects of initial conditions on the model forecasts. Such residual dependence would be expected to be independent of the choice of hydrological model used to translate the independent precipitation series to streamflow.

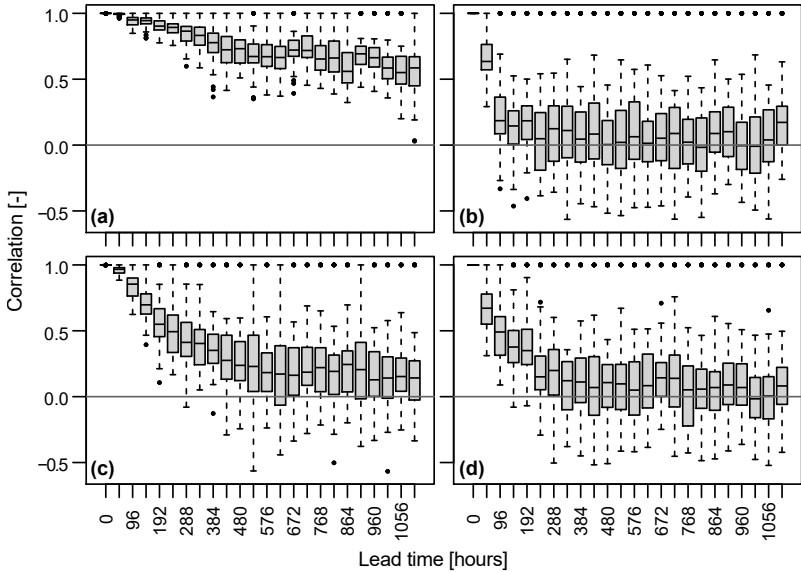

**Figure 5.** Member (in)dependence (Spearman's correlation) per lead time (0 to 1104h (46 days)) across the 10 perturbed ensemble members for four example stations with different flood seasonality ratios (strong summer vs. strong winter when going from upper left to lower right): (a) Kemijoki, Ounasjoki (strong summer flood regime), (b) Osterach, Reckenberg (summer flood regime), (c) Rhine, Maxau (winter flood regime), and (d) Jouanne, Force (strong winter flood regime) (Figure 1).

We seek to better understand which types of catchments show high/low ensemble member dependence across lead times. Therefore, we compute median Spearman's rank correlation across the 10 ensemble members and 24 lead times for each of the 234 catchments and try to relate this median correlation to a catchment's flood seasonality ratio. The flood seasonality ratio $R_F$ is computed as $R_F = Q_{95s}/Q_{95w}$, where $Q_{95s}$ represents $Q_{95}$ in summer (months: April–Sept) and $Q_{95w}$ represents $Q_{95}$ in winter (months: Oct–March). $R_F > 1$ and $< 1$ represent catchments with more severe summer than winter floods and more 235   severe winter than summer floods, respectively. We also considered other metrics to explain median lead time correlation such as the baseflow index (Ladson et al., 2013) or catchment area but did not find any meaningful relationships.





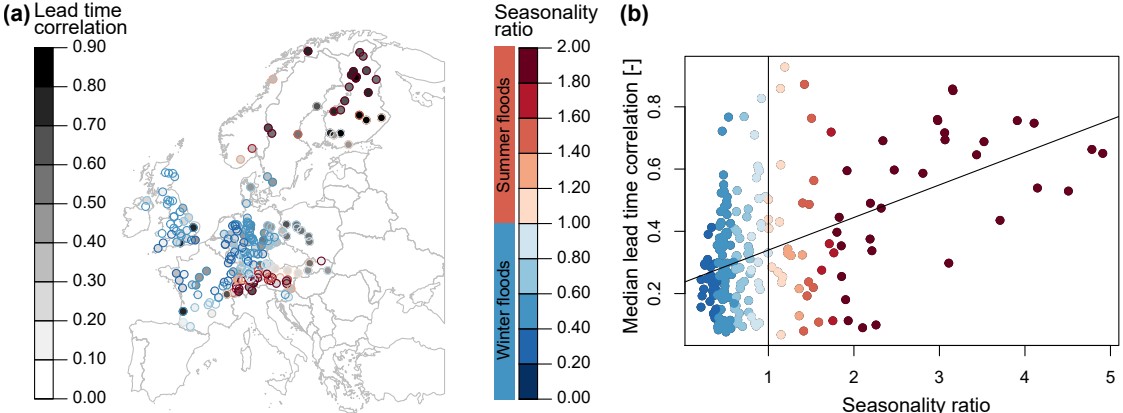

**Figure 6.** Median annual maxima dependence across all lead times and ensemble members per catchment: (a) spatial variation in median correlation (grey outlines) and seasonality ratio where red colors indicate higher floods ($Q_{95}$) in summer than winter and blue colors indicate higher floods in winter than summer; (b) relationship between median correlation and seasonality ratio. The vertical black line indicates the transition from winter to summer dominated flood regimes and the trend line was derived using linear regression.

Median lead time dependence shows clear spatial patterns with higher dependencies in the Alps and Scandinavia than in the rest of Europe (Figure 6a). These regions with higher median dependencies are characterized by a summer flood regime as they are partly influenced by snowmelt contributions (Berghuijs et al., 2019). Median lead time correlation seems to generally

increase with higher seasonality ratios (Figure 6b), i.e. the more summer-/snow-dominated a flood regime is, the higher the AM dependence. However, some of the winter-/precipitation- dominated regimes can also have high dependence values.

The high dependence at low lead times suggests that simulations at lower lead times should be removed before pooling flood events for frequency analysis. In order to determine the lead times to be excluded, we compute median AM dependence across ensemble members and catchments for each lead time and perform a Pettitt change point test (Ryberg et al., 2019) on

the resulting median time series (Figure 7a). The change point analysis suggests that dependence values stabilize on average at around 528 hours, i.e. 22 days. As an alternative to using a single threshold for all catchments, one could use a variable threshold, which is lower for catchments with lower dependence values and higher for catchments with higher dependence values. We decided to work with one single threshold for simplicity. The implementation of a 22 day independence threshold compares well with independence thresholds identified and used in previous studies applying reforecast pooling (10 days,

Breivik et al. (2014); 30 days, Kelder et al. (2020)).





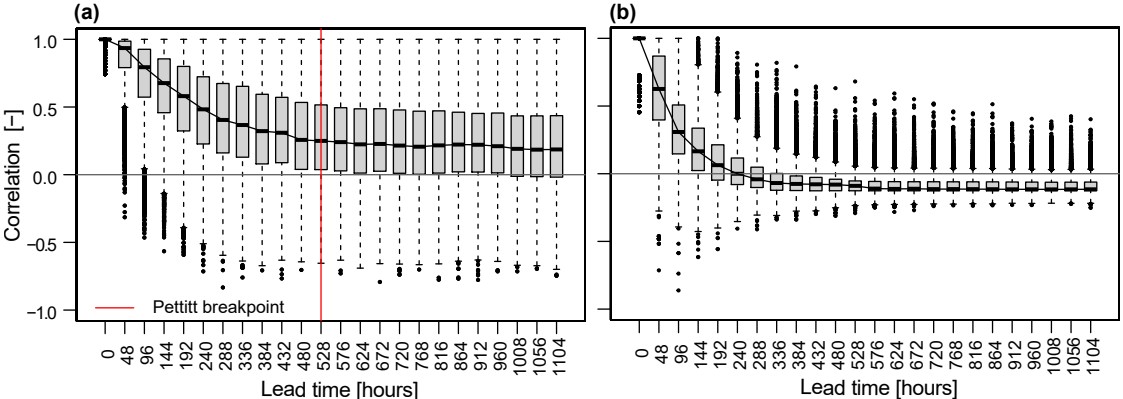

**Figure 7.** Median Spearman's correlation across ensemble members and catchments per lead time (0 to 1104 hours) for: (a) annual maxima and (b) peak-over-threshold events. The median correlation per lead time is indicated by the grey line and the break point in this median series derived by the Pettitt test by the red line. No red line is shown in (b) as the break point was determined using the AM series shown in (a).

Our flood frequency analysis therefore pools flood events derived from streamflow time series of the 10 perturbed members for each lead time > 22 days. Such pooling allows us to substantially increase sample size (i.e. 130 times; 13 lead times and 10 perturbed runs). To further reduce dependence, our analysis relies on peak-over-threshold instead of annual maxima events (Figure 7b), which substantially reduces dependence at all lead times if we compute Spearman's rank correlation for
exceedance time series where non-exceedances are replaced by 0 (Breivik et al., 2013).

## 3.3 Flood frequency analysis

Flood estimates derived by theoretical distributions fitted to pooled peak-over-threshold (POT) flood events from 10 ensemble members and 13 lead times are more robust, i.e. have smaller uncertainty, than flood estimates derived from distributions fitted to a small sample of observed POT events, as illustrated in Figure 8 for four example catchments (Figure 1). Observation- and
simulation-based estimates do not just differ in terms of uncertainty but also in terms of the magnitudes of the best estimates.





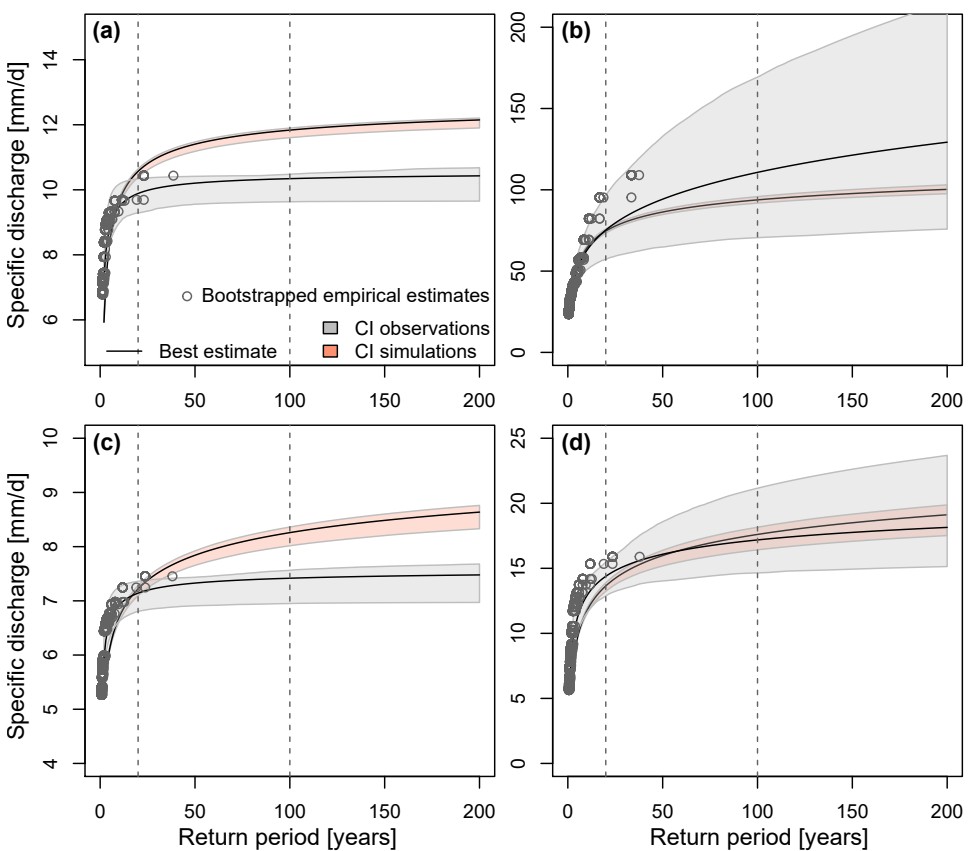

**Figure 8.** Observed vs. simulated flood frequency curves including uncertainty bounds for four example catchments with different seasonality ratios: (a) Kemijoki, Ounasjoki, (b) Osterach, Reckenberg, (c) Rhine, Maxau, and (d) Jouanne, Force. The observed and simulated best estimate frequency curves are indicated by black lines, the corresponding 90% confidence intervals by shaded polygons, and bootstrapped empirical return period estimates by grey dots. Confidence intervals are derived using bootrapping.

The differences between observation- and simulation-based best estimates and uncertainty ranges vary by return period and by catchment (Figure 9). Relative differences between observation- and simulation-derived best estimates are mostly positive, i.e. observed quantiles tend to be larger than simulated quantiles (Figure 9a). These relative differences increase with return period length. Similarly, observation-derived uncertainty ranges are wider than simulation-derived uncertainty ranges and these

differences also increase with return period length (Figure 9b). Both the relative differences in best estimates and uncertainty bounds depend on catchment area and elevation to some degree (Figure 9c, d). Low-elevation and large catchments generally show lower relative differences in best estimates and uncertainty than high-elevation and small catchments. As the effect of area on relative differences is stronger than the effect of elevation, there are no clear spatial patterns in relative differences between observed and simulated best estimates and uncertainty bounds. Please also note that the relative differences between simulated

and observed best estimates and uncertainty bounds are independent of model performance, which means that increasing the





cut-off threshold for EFAS model performance does not necessarily lead to an increase in the similarity between observed and simulated flood estimates.

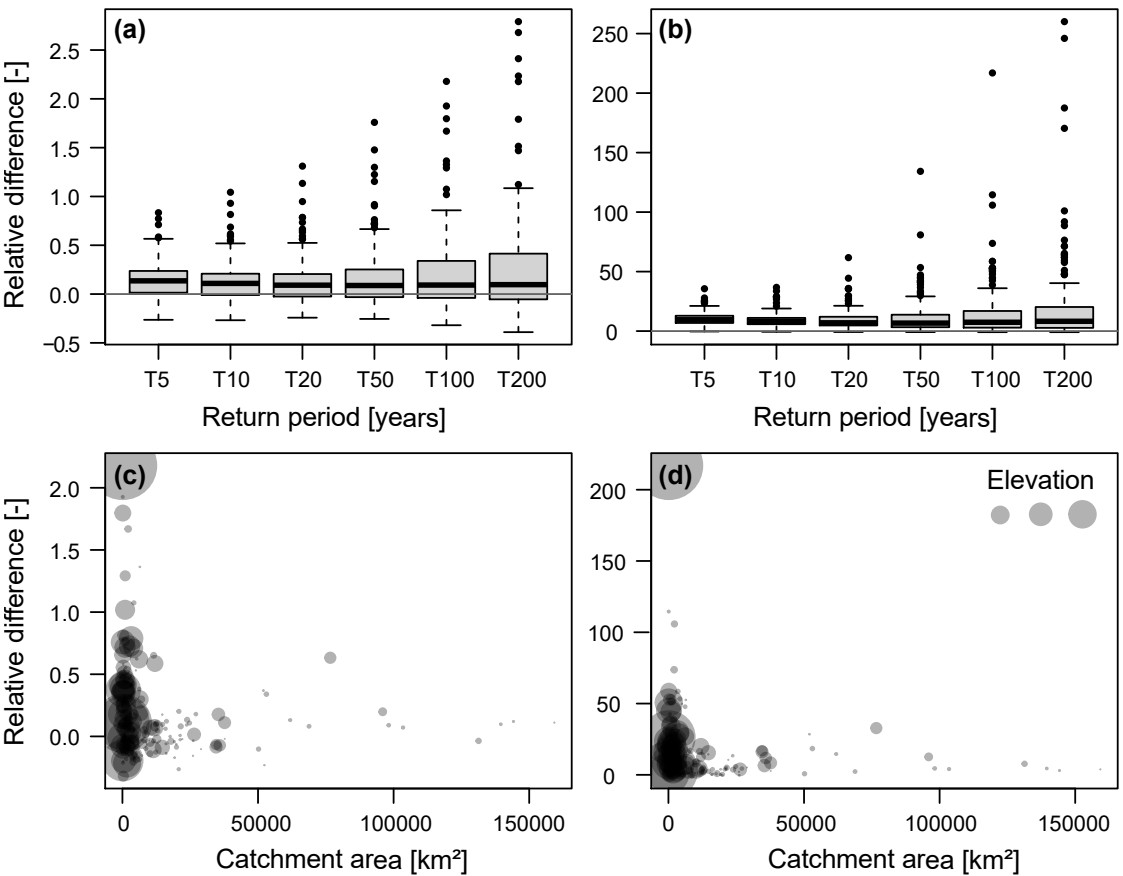

**Figure 9.** Relative difference between observed and simulated ((obs-sim)/sim) (a) best estimates and (b) range of uncertainty bounds ($Q_{95} - Q_{05}$) for different return periods (5, 10, 20, 50, 100, and 200 years) across catchments (1 point in boxplot corresponds to one catchment). Relative difference between observed and simulated ((obs-sim)/sim) (c) best estimates and (d) range of uncertainty bounds ($Q_{95} - Q_{05}$) for the 100-year return period in relation to catchment area and elevation. The larger the dot is, the higher elevation of a catchment is.

We now use the best estimates derived by ensemble pooling to map spatial patterns of flood quantiles over Central Europe for different return periods (Figure 10). Flood quantiles are highest in the Alps and Great Britain and lowest in Northern Germany and Scandinavia independent of the return period. These spatial patterns corroborate previous findings that the Alps and Great Britain are regions with a comparably high number of flood events per year (Mangini et al., 2018) and that observation-based 100-year specific discharge is highest in the Alps, Great Britain and Norway and lowest along the Atlantic coast (Blöschl et al., 2019). Additionally, we find that the local flood quantiles are positively related to mean slope and mean precipitation of a catchment, and negatively related to the number of dams and mean snowmelt of a catchment (Figure 11). Catchments with





steep slopes and higher mean precipitation tend to have higher flood quantiles, while catchments with a greater numbers of dams and higher snowmelt contributions tend to have smaller flood quantiles.

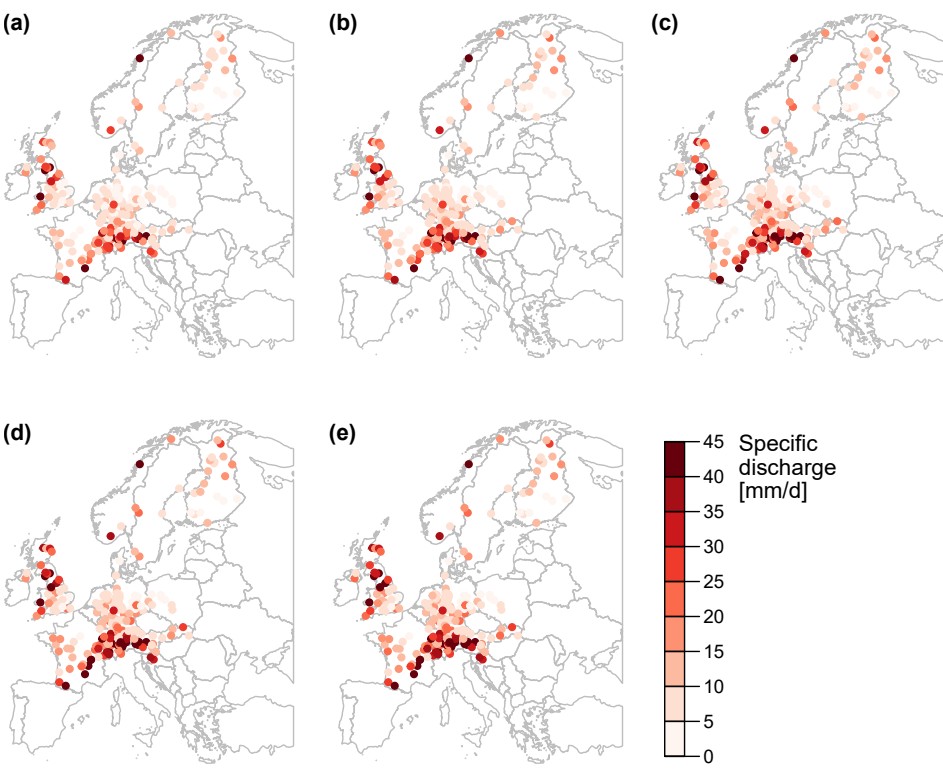

**Figure 10.** Theoretical flood quantiles corresponding to return periods of (a) 10, (b) 20, (c) 50, (c) 100, and (e) 200 years derived from pooling POT events extracted from time series simulated for 10 ensemble members and 13 lead times ($\geq$ 528 hours, sample size = 2600 (13 lead times $\times$ 10 members $\times$ 20 years)).





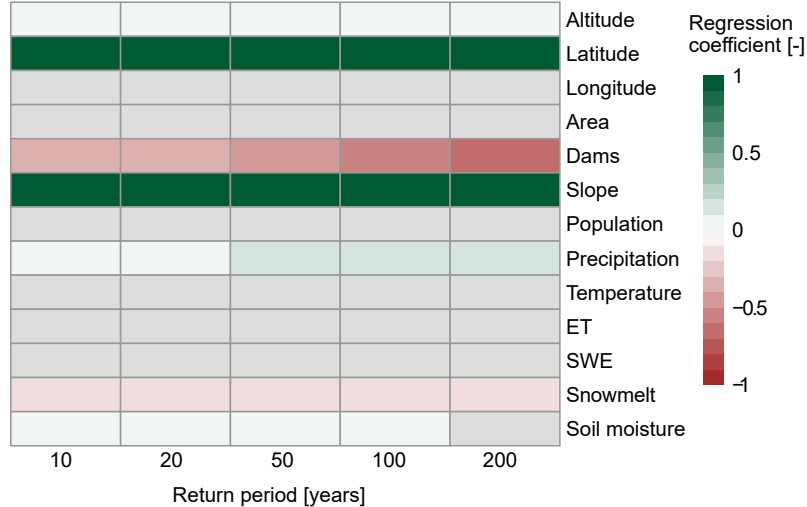

**Figure 11.** Predictor importance for flood quantiles. Regression coefficients for significant explanatory variables retained when choosing the linear model with the lowest BIC ($\alpha = 0.05$). Green and red colors indicate positive and negative relationships between flood estimates and catchment characteristics, respectively. Grey colors indicate non-included explanatory variables.

Ensemble pooling can also be used to derive regional flood estimates, i.e. to compute the probability that a certain percentage of catchments within a region, i.e. large river basin, are jointly flooded (Figure 12). Regional floods with a 30% coverage, i.e. floods affecting at least 30% of catchments within a region, occur relatively frequently (return periods < 10 years) both in Central Europe as well as Scandinavia (Figure 12a). In contrast, regional events with a 50% coverage are more likely in Central Europe (lighter colors) than in Scandinavia (darker colors). Very widespread events with 70% spatial coverage become very rare (return periods > 90 years) in most parts of Europe except in the Weser, Elbe, and Oder river basins.

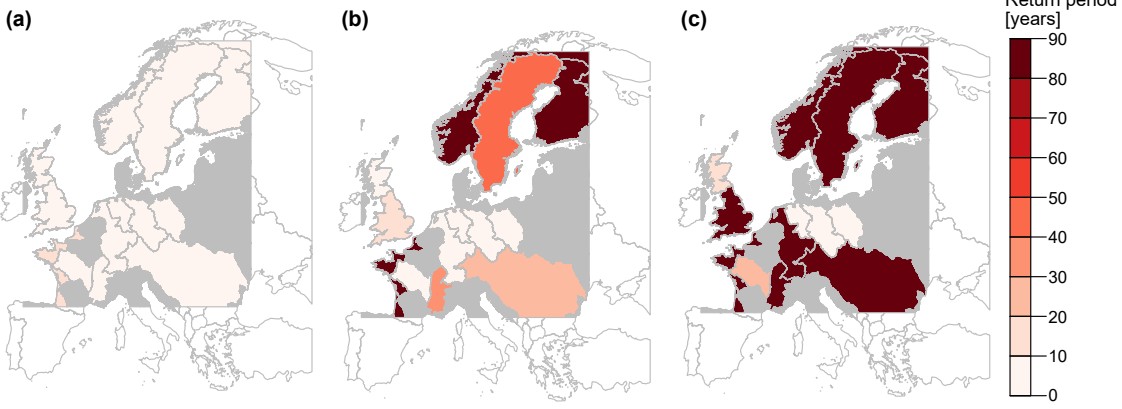

**Figure 12.** Probabilities of regional flooding for European river basins with more than five catchments: (a) 30% affected, (b) 50% affected, and (c) 70% affected. Regions with less than 5 catchments where regional flood probabilities could not be determined are highlighted in grey and regions not covered by our dataset displayed in white.





## 4 Discussion

Pooling flood events derived from a streamflow reforecast ensemble substantially increases the size of the sample available for
flood frequency analysis. In doing so, it enables the study of very rare extremes absent in relatively short observed time series.
Increasing the sample size also facilitates the study of spatial patterns in the distribution of flood estimates corresponding to
long return periods (e.g. 200 years; Figure 10e) and notably reduces uncertainty in most cases (Figure 9b) independent of
the original EFAS model performance. Furthermore, it enables the study of rare spatial extremes, i.e. events that may affect
multiple catchments at once (Figure 12).

However, the utility of reforecast pooling rests on the performance of the underlying hydrological simulations. The use
of reforecast simulations instead of observations comes at the cost of potentially introducing uncertainty through simulated
meteorological input or the hydrological model itself (structure and parameters) (Clark et al., 2016). These uncertainties may
result in biased simulations, which may either under- or overestimate the whole or specific parts of the streamflow distribution.
Such bias can be partly reduced by using bias correction techniques such as quantile mapping (Figure 2). Yet the the plausibility
of unprecedented extremes relies on the realism of hydrological simulation in the model system, i.e. a reliable representation of
hydrological processes and their drivers. Further improvement in model representation of high flows may be needed to reduce
bias and improve process representation (Mizukami et al., 2019; Brunner et al., 2021).

An additional limitation is the spatial applicability of the approach. As hydrological model simulations must be bias cor-
rected, the use of ensemble pooling is currently limited to catchments for which streamflow observations are available. This
requirement limits the application of the pooling approach to gauged catchments. In theory, using simulations instead of ob-
servations would enable extension of the spatial coverage to ungauged locations. However, such an extension would only be
possible if no bias correction was required or if bias correction could be regionalized and applied to all catchments.

Sample size is only effectively increased compared to observations if the simulated flood samples for different ensemble
members can be considered independent (Kelder et al., 2020). However, such independence is more difficult to achieve in
hydrological than meteorological systems as hydrological systems exhibit substantial memory effects e.g. through snow or soil
moisture storage (Berghuijs et al., 2019; Brunner et al., 2020a). These memory effects introduce varying degrees of dependence
to ensembles of simulated annual maxima time series (Figure 5). The dependence is highest in catchments with high seasonality
and where floods predominantly occur in summer under the potential influence of snowmelt (Figure 6). Dependence does not
depend strongly on variables which typically affect streamflow persistence such as catchment area or baseflow index. Still, any
such dependence can be notably decreased if annual maxima events are replaced by peak-over-threshold events. Using POT
events has the advantage that besides event magnitudes and timing, the number of events may also vary. This approach means
that a one-to-one relationship between events extracted from two different ensemble members can no longer be established.

The flood ensemble pooling approach described herein is not limited to the EFAS reforecasts over Europe but could also
be applied to other streamflow reforecast modeling systems such as the Global Flood Awareness System (GloFAS; Alfieri
et al., 2013) or the Global Flood Forecasting Information System (GLOFFIS; Emerton et al., 2016). Moreover, the pooling
approach may be beneficial to other types of hydrological extremes beyond flood frequency analysis, such as droughts. Such an





extension would require model evaluation targeted at the variable of interest. Hydrological extremes extracted from streamflow reforecasts may also be used in combination with climatic extremes extracted from meteorological reforecasts to study the frequency of compound events (such as joint pluvial and fluvial flooding), or the drivers of various extremes. In any case, the

use of simulated extremes pooling requires careful model evaluation and is likely to require some form of bias correction to ensure the fidelity of extremes.

## 5 Conclusions

Pooling of publicly-accessible reforecast flood events such as those generated through the European Flood Awareness System (EFAS) can be a useful tool to improve the robustness of flood estimates, particularly for rare events with long return peri-
ods. However, as with other extremes (Kelder et al., 2020), such pooling is only effective if simulated floods show little bias and model drift across lead times, and if the floods extracted from different ensemble members are sufficiently independent to increase the effective sample size available for frequency analysis. Bias can be removed using bias correction techniques such as quantile mapping. The degree of dependence is subject to the catchment (with summer-flood-dominated catchments showing higher dependencies than winter-flood-dominated catchments), lead time (with decreasing dependence at longer lead
times), and event type (with peak-over-threshold events showing lower dependence than annual maxima events). The higher dependence of summer-flood-dominated catchments than winter-flood-dominated catchments suggests that catchment memory through snow storage effects is an important determinant of dependence and that catchments with a more predictable seasonality may have greater member dependence. We recommend pooling peak-over-threshold events from ensemble runs generated for lead times > 22 days, because floods are less dependent on average beyond such a lead time, even when using annual
maxima events, and because dependence is lower for POT than AM events.

Our application of the pooling approach over 234 European catchments shows that local floods are most extreme in the Alps and Great Britain and least extreme in Scandinavia and Central Europe. It also indicates that regional extreme flood events, in which a large fraction of catchments flood simultaneously, are more likely in Central Europe than in Scandinavia. We conclude that pooled reforecast ensembles are beneficial in studying the probability of extreme and spatially extensive events in the
case of accurate model representation of hydrologic extremes, as they help provide flood estimates with considerably reduced uncertainty compared to observation-derived flood estimates.

*Data availability.* The reforecasts of river discharge generated through EFAS are available for download through the Copernicus data store: https://cds.climate.copernicus.eu/cdsapp#!/dataset/efas-reforecast?tab=overview and observed discharge through the GRDC: https://www.bafg.de/GRDC/EN/02_srvcs/21_tmsrs/riverdischarge_node.html.

*Author contributions.* MIB and LS jointly designed the study and developed the methodology. MIB prepared the data, performed the analyses, and wrote the first draft of the manuscript. LS revised and edited the manuscript.





*Competing interests.* The authors declare no competing interests.

*Acknowledgements.* This work was supported by the Swiss National Science Foundation via a PostDoc.Mobility grant (Number: P400P2_183844,
granted to MIB) and a John Fell Fund grant (to LJS). We would like to acknowledge high-performance computing support from Cheyenne
(doi:10.5065/D6RX99HX) provided by NCAR's Computational and Information Systems Laboratory, sponsored by the National Science
Foundation.





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
