# Peer review of "Extreme floods in Europe: going beyond observations using reforecast ensemble pooling"

_Hydrology and Earth System Sciences, 2021_

## Author Comment (AC1)

The paper of Brunner and Slater demonstrates the utility of a technique based on reforecast ensemble pooling to assess the frequency of extreme floods. Starting from selected catchments available in the EFAS database, they apply this technique in central northern Europe, deriving several outcomes concerning also the possibility of regional flooding.

The paper is generally well written and provides useful insights for the application of the reforecast pooling (or UNSEEN) approach to flood frequency analysis (the authors claim that this is the first application with such a variable). In the Discussion section, the limits of the methodology are also clearly outlined (the main being, as usual, the availability of observed streamflow data). However, the paper could be made clearer and more straightforward. I have two main comments.

My first comment concerns the question: why should one prefer reforecast pooling to other methods such as "classical" stochastic simulations? This question is fundamental to highlight the utility of the proposed method. It should be considered both in the Introduction (which could be enlarged) and Discussion sections.
**Reply:** *Thank you for pointing out the need to better discuss the relationship of reforecast pooling with other methods available for increasing sample size. We did not intend to suggest that reforecast pooling is to be preferred over other methods. Rather, we wanted to present it as an alternative to existing methods such as stochastic simulation or large climate ensembles. To clarify this, we substantially reworked and extended the introduction and added a short paragraph to the discussion section, where we discuss how the reforecast ensemble pooling method relates to other methods: 'Therefore, streamflow reforecast ensemble pooling represents a suitable alternative to stochastic or climate model large ensemble approaches for studying the frequency and magnitude of rare extreme events. Similar to large ensemble approaches but in contrast to stochastic approaches, reforecast-based simulation approaches rely on physical representations of the hydrological cycle. Such physical representation may be especially valuable if relationships between different variables are of interest and if one wishes to study the physical drivers of flood events. In contrast, stochastic models have the advantage of being relatively straightforward to implement and are potentially less computationally intense.'*

My second main comment concerns methodology, especially Sections 2.2 and 2.3. Applying the reforecast pooling technique does not look very straightforward, given that several preliminary steps are needed. Another aspect that made it difficult for me understanding these sections, which I had to reread several times, is the sudden description of operations that had not been introduced previously. E.g., the need for bias correction (L124) comes abruptly, such as the use of linear regression models (L166). I suggest introducing better the different steps, linking them to specific objectives, possibly aided by a flowchart. Please find below some specific comments. I hope my review can help to improve the quality of the paper.
**Reply:** *Thank you for pointing out that the methods section needed greater clarity. We adopted your suggestion and designed a flow chart to visualize links between the different working steps. We also readjusted the order in which the different working steps are presented by reorganizing the methods section in a chronological way.*

LL12-13: "… specific flood return levels are highest in …": not very clear. I would better tell that, given a return period, specific floods are higher in steep and wet regions etc.
**Reply:** *Thank you very much for this rephrasing suggestion, which we adopted.*

L49: are they mean elevations?
**Reply:** *Yes, we specified that we are talking about mean elevations.*

L54: what does "acceptable" mean in this context? Maybe some details could be disclosed here
**Reply:** *We added the model evaluation criteria introduced in Section 2.2 to the introductory paragraph of the Methods and Materials section.*

Section 2.2: as I wrote before, a flowchart would help a lot
**Reply:** *Thank you very much for this great suggestion, which we adopted by adding a new Figure 1, a flow chart indicating the most important analysis steps and links between them.*

Fig. 2: it looks like not always quantile mapping produces better results (e.g. fig. 2b). Could the authors provide more details about the overall analysis? However, please change colours. Shades of red are too similar.
**Reply:** *It is correct that median correction may perform as well or even slightly better than quantile mapping for certain parts of the distribution as illustrated in Figure 2b (now 3b). However, we still found that overall, quantile mapping resulted in more satisfactory results. The main reason for not just correcting by the median was because our study focuses on extreme flows, which can be corrected by applying quantile mapping. We changed the colors to make the different lines easier to distinguish.*

Fig.3 is a bit minimalistic. I'm confident it can be improved.
**Reply:** *The main idea behind the figure was to illustrate which part of the dataset was used for the frequency analysis and which part was excluded. As part of the information previously included in Figure 3 is now included in the new Figure 1 (10 perturbed members, 24 lead times, overall sample size), we removed the previous Figure 3.*

L180: the the
**Reply:** *We removed the redundant 'the'*

L188: r%?
**Reply:** *We corrected this to 'p%'*

Fig. 9(c) and (d) are not very clear. Maybe the x-axis could have a log scale.
**Reply:** *Thank you very much for this great suggestion, which we adopted and helped to improve the figure.*

Fig. 10: please correct the typo in the caption ")". Furthermore, the five maps are very similar. Maybe some of them could be removed.
**Reply:** *We replaced the inner pair of brackets by '[]'. It is true that the spatial patterns shown on the five maps look quite similar but we would prefer to retain all of them because the different return periods are again picked up in Figure 11.*

L280: what about the results concerning latitude? In Fig. 11 it is dark green for all return periods. Please check. Anyway, results concerning the other variables (i.e., slopes, mean precipitation, dams and snowmelt) are quite obvious and make the analysis less interesting.
**Reply:** *We agree that it is difficult to argue why latitude should physically be an important predictor of flood magnitudes. We therefore excluded longitude and latitude as potential predictors from the analysis and redid figure 11. We now see a relatively strong negative relationship between temperature and flood magnitude. That is, higher flood magnitudes for catchments with colder climates (e.g. those in the Alps).*

L299: the the
**Reply:** *We removed the redundant 'the'*

L314: "…any such…" please check
**Reply:** *We removed 'and'*

---

## Author Comment (AC2)

**Peter Salamon Referee #2**

**General comment:**

The manuscript presents an approach to increase the sample size for the estimation of the frequency of flood events. The approach is based on pooling of reforecast ensemble members and has not been previously assessed for flood frequency analysis.

The paper is overall well written and structured. The approach presented in this manuscript is of high interest as estimating flood frequency in practice is often hampered by short observational records. The discussion section outlines the possible limitations of the approach. My main concern is related to the data used for the study. The study uses EFAS v3.0 historical simulations to assess whether the selected stations have a good performance when comparing simulations and observations. However, the EFAS reforecast data set used for the ensemble pooling is based on EFAS v4.0 which includes a completely new model calibration, upgrades to static fields for the hydrological model LISFLOOD and a change from a daily timestep to a 6 hourly timestep. Overall, EFAS model performance from v3.0 to v4.0 has increased significantly and therefore it is not recommended to select stations based on v3.0 and perform an analysis using reforecasts that are based on EFAS 4.0. As this has an impact on all results and analysis in the manuscript a major revision is required.

**Main concern:**

As described in the general comment EFAS reforecasts are based on EFAS v4.0 as is also indicated in the metadata on the Climate Data Store (https://cds.climate.copernicus.eu/cdsapp#!/dataset/efas-reforecast?tab=overview ). However, EFAS historical simulations v3.0 were used to pre-select stations with a good fit between simulated and observed discharge. Given that EFAS 4.0 contains a completely new model calibration with more calibration stations (1137 for v4.0 instead of 717 stations for previous EFAS versions), and upgrades to static fields for the hydrological model LISFLOOD and a change from a daily timestep to a 6 hourly timestep as is described in detail in the EFAS wiki (see here: https://confluence.ecmwf.int/display/COPSRV/EFAS+v4.0 ) it is not recommendable to use EFAS v3.0 model performance to pre-select stations and then use those pre-selected stations with EFAS re-forecasts from EFAS v4.0.
**Reply:** *Thank you for highlighting that there were inconsistencies between the description of the calibration procedure we provided (which implied we were using EFAS 3.0) and the actual use of EFAS 4.0 (which we employed for both our station selection and analysis). In the analysis, we did compare EFAS 4.0 historical runs with observed GRDC data for model evaluation, i.e. the EFAS version used was consistent. However, our initial description of the calibration procedure suggested that we were using runs from EFAS 3.0 to select pre-stations. This is not the case and we updated the description of the calibration procedure to match the calibration procedure used in EFAS 4.0 as documented on the EFAS Wiki pages.*

Furthermore, the authors do not describe in detail how the simulated data was extracted from the EFAS simulations. EFAS output has a spatial resolution of 5km x 5km. The coarse spatial resolution of the hydrological model LISFLOOD used in EFAS requires an upscaling of the river drainage network from a high resolution dataset to the 5km x 5km grid scale. This means that coordinates of gauging stations cannot be used directly to extract simulated timeseries of discharge from the EFAS simulations as original gauging station coordinates may be located just next to the main river channel on the coarse grid scale. Instead, before extracting simulated time series it has to be checked whether the drainage area of the EFAS grid pixel corresponds to the drainage area as provided by the data provider (here GRDC). While smaller differences in the drainage area are expected due to the different spatial scales, if there is a large difference, it means that coordinates have to be shifted to ensure an adequate match. For this purpose the drainage area of the LISFLOOD/EFAS network is available on the C3S CDS (https://cds.climate.copernicus.eu/cdsapp#!/dataset/efas-historical?tab=overview ). This is especially important for gauging stations with very small drainage areas which seem to have been used predominantly in this study (Fig.

9). Furthermore, LISFLOOD simulates lakes and reservoirs as points on the channel network. It is not recommended to extract simulated time series at the same pixel where the reservoir or lake is located but to either extract the time series on the upstream or downstream pixels of lakes and reservoirs (depending on the location of the gauging stations for observations). More info can be found on the model documentation of LISFLOOD (https://ec-jrc.github.io/lisflood/ ). The location of lakes and reservoirs on the EFAS grid can be found also on the EFAS map viewer (https://www.efas.eu/efas_frontend/#/home ) .

*Reply: Thank you for highlighting the need to (1) clarify the method we employed to match the observational GRDC sites with the corresponding EFAS grid cells; and (2) assess the correspondence between the catchment area of the GRDC sites and the upstream area of the corresponding EFAS grid cells.*

*As we were working with a large data set to start with (>1500 GRDC catchments), we were not able to manually identify EFAS pixels for each of the GRDC stations in the initial data pool. Because manual matching seemed infeasible for such a large data set, instead we identified one grid cell per GRDC catchment using latitude-longitude (or coordinate) matching. As indicated by the reviewer, not all of these pixels may necessarily correspond to the 'correct' pixel with the same upstream area as the GRDC catchment. To avoid including catchments with a mismatch between upstream pixel area and GRDC catchment area, we have now pre-filtered the catchments and only included those catchments which showed a relative difference in catchment area between upstream pixel area and GRDC catchment area of < 20% in the initial catchment pool. Using this dataset (that only included catchments with a good area match), we then applied the model evaluation process, which aimed to filter out any additional catchments where simulation performance with respect to high flows was not considered sufficient for our flood frequency analysis. This two-step process (area correspondence verification and performance evaluation) allowed us to select a data set of 234 clearly-located catchments with good model performance in terms of high flows. We updated the respective passages in the manuscript to clarify the two-step procedure.*

Finally, we have found several data quality issues with the observed discharge data in GRDC in the past. We recommend strongly to have at least a visual check of the observed data that is selected for the analysis.
*Reply: Thank you for pointing out the need to quality check observed discharge data downloaded from the GRDC. We visualized both the observed and simulated time series of the 234 catchments we used for the frequency analysis and did not detect any obvious inconsistencies in the observed data.*

**Minor comments:**

Chapter 2.1, page 3: EFAS 4.0 as well as EFAS3.0 have been calibrated using Kling Gupta efficiency and not NSE. For more details on the EFAS versioning and what changes are included in each EFAS version please see here for a detailed description: https://confluence.ecmwf.int/display/COPSRV/EFAS+versioning+system  The reference to Smith et al. refers to previous and outdated EFAS model versions that is not available on the C3S CDS.
*Reply: Thank you for highlighting that the calibration procedure described in Smith et al. does not refer to the most recent calibration setup. We updated the description of the calibration procedure following information provided in the EFAS Wiki (https://confluence.ecmwf.int/display/COPSRV/Modelling+upgrade+for+EFAS+v4.0).*

Chapter 2.2, page 5, line 109: "…model stability…": Please describe what is meant here with model stability? In terms of what?
*Reply: We have rephrased the sentence to explain what we mean by model stability: 'Next, we assess the suitability of the perturbed ensemble streamflow simulations for ensemble pooling by evaluating whether individual simulation runs can be considered independent and whether the model is stable, i.e. simulated distributions are stable across lead times (Kelder et al. 2020).'*

Chapter 2.2, page 5: It is stated that Spearmans rank correlation can only be computed for AM and not directly for POT (lines 114-116). However, in the following sentence you write that you calculated Spearmans correlation for POT. Please clarify!

**Reply:** *Thank you for pointing out the need for clarification. We rephrased the section to: 'Note that such correlation can directly only be computed for AM and not for peak-over-threshold (POT) series because POT series may differ across ensemble members in the number of events chosen for analysis and not just in timing and magnitude. It can therefore be assumed that the POT events used in our subsequent analyses are more independent than AM events. To illustrate this, we indirectly compute Spearman's correlation for pairs of POT time series by using events where at least one of the time series exceeds a threshold and by replacing non-exceedances in the other time series by 0 (not ideal because this might artificially introduce some sort of dependence).*

Chapter 2.3, page 8, line 180: … the the…. Please correct.
**Reply:** *We corrected this typo.*

Chapter 3.1, page 8: This is a repetition of Chapter 2.2. Please remove Chapter 3.1!
**Reply:** *Good point, we removed Section 3.1.*

Chapter 3.3, page 13, lines 269-271: Please describe the evidence for claiming that relative differences between simulated and observed best estimates and uncertainty bounds are independent of model performance.
**Reply:** *We specified that by 'independent', we mean 'uncorrelated'. That is, we computed the correlation between relative differences of simulated and observed best estimates and different model performance metrics (e.g. KGE).*

Chapter 3.3, Fig. 11: I disagree with your statement that flood quantiles are positively related to mean precipitation. According to Fig. 11 there is only a very weak positive relation.
**Reply:** *It is correct that the regression coefficient for precipitation is weaker than the one for slope or latitude but the explanatory variable is still significant and positively related to flood magnitude. We therefore think that this statement is correct.*

Chapter 3.3, Fig. 11: Please explain why there is such a strong positive relation to Latitude according to Fig. 11? This is not mentioned at all in the text.
**Reply:** *We agree that it is difficult to argue why latitude should physically be an important predictor of flood magnitudes. We therefore excluded longitude and latitude as potential predictors and redid figure 11. We now see a relatively strong relationship between temperature and flood magnitude. That is, higher flood magnitudes for catchments with colder climates (e.g. those in the Alps).*